# Interactions between Gut Microbiota and Oral Antihyperglycemic Drugs: A Systematic Review

**DOI:** 10.3390/ijms25063540

**Published:** 2024-03-21

**Authors:** Nicoleta Mihaela Mindrescu, Cristian Guja, Viorel Jinga, Sorina Ispas, Antoanela Curici, Andreea Nelson Twakor, Anca Mihaela Pantea Stoian

**Affiliations:** 1Department of Diabetes, Nutrition and Metabolic Diseases, “Carol Davila” University of Medicine and Pharmacy, 050474 Bucharest, Romania; nicoleta-mihaela.mindrescu@drd.umfcd.ro (N.M.M.); cristian.guja@umfcd.ro (C.G.); viorel.jinga@umfcd.ro (V.J.); anca.stoian@umfcd.ro (A.M.P.S.); 2National Institute of Diabetes, Nutrition and Metabolic Diseases “NC Paulescu”, 030167 Bucharest, Romania; 3Clinical Hospital, “Prof. Dr. Th. Burghele”, 061344 Bucharest, Romania; 4Department of Anatomy, Faculty of General Medicine, “Ovidius” University, 900470 Constanta, Romania; sorina.ispas@365.univ-ovidius.ro; 5Department of Cellular and Molecular Biology, and Histology, “Carol Davila” University of Medicine and Pharmacy, 050474 Bucharest, Romania; antoanela.curici@umfcd.ro; 6Department of Internal Medicine, Emergency County Hospital, 900591 Constanta, Romania

**Keywords:** oral antihyperglycemic drugs, gut microbiota, diabetes mellitus

## Abstract

The intestinal microbiota refers to the collection of microorganisms that exist in the human gut. It has been said that bacteria influence the development of metabolic diseases, such as diabetes mellitus, as they have roles in immunomodulation, protection against pathogens, blood vessel growth, repairing the intestinal wall, and the development of the neurological system. In this review, we look at the latest research regarding interactions between gut microbiota and oral antihyperglycemic drugs and we present data suggesting that the microbiome may help counteract the reduced glucose tolerance and insulin resistance associated with metabolic disorders. We found that antidiabetic drugs can have significant impacts on gut microbiota composition and function, potentially influencing both the efficacy and side effects of these medications. Additionally, we discovered that microbial-based therapeutics, including probiotics, prebiotics, and postbiotics, and fecal microbiota can be considered when discussing preventive measures and personalized treatment options for type 2 diabetes mellitus. Understanding how antidiabetic drugs modulate gut microbiota composition and function is essential for optimizing their therapeutic efficacy and minimizing potential adverse effects. The relationship between the gut microbiota and glycemic agents, not fully understood, is currently the subject of increasing research and discussion. It has been proven that the microbiome can impact the effectiveness of the medications, but further research in this field may uncover novel therapeutic strategies for diabetes and other metabolic disorders by targeting the gut microbiota.

## 1. Introduction

Diabetes mellitus is a prevalent and persistent metabolic illness affecting, in 2021, around 537 million individuals globally [1]. By 2045, International Diabetes Federation projections show that one in eight adults, approximately 783 million, will be living with diabetes—an increase of 46% [2]. Type 2 diabetes mellitus (T2DM) is influenced by a combination of genetic, environmental, and lifestyle factors [3]. Excess adiposity, particularly central obesity (abdominal fat), is one of the strongest risk factors for T2DM [4]. Dietary patterns, such as processed foods, refined carbohydrates, sugary beverages, and saturated fats, contribute to obesity, insulin resistance, and dyslipidemia [5]. The relationship between changes in gut microbiota and the development of T2DM is complex and multifaceted. While a definitive cause-and-effect relationship has not been fully established, accumulating evidence suggests that alterations in gut microbiota composition and function may contribute to the pathogenesis of T2DM through various mechanisms. [6]. Existing research is currently investigating the role of gut microbiota as a biomarker for type 2 diabetes mellitus and a potential therapeutic approach for treating the disease [7].

The gut microbiota refers to the collection of microorganisms that belong to the gastrointestinal tract (GI). The gut is host to a vast number of bacteria, exceeding 100 trillion, with a significant concentration in the colon [8]. Bacteria are taxonomically categorized from the species down to the phylum (such as *Firmicutes* and *Bacteroidetes*, which are the main ones) [9]. Other phyla are *Actinobacteria*, *Proteobacteria*, and *Cerrucomicrobia* [10]. Research led by LeBlanc et al. has shown that the gut microbiota in humans may produce vitamin K and many water-soluble B vitamins, including biotin, cobalamin, folates, nicotinic acid, pantothenic acid, pyridoxine, riboflavin, and thiamine [11]. Gut microbiota may be altered by antidiabetic medicines and, in turn, impact an individual’s response to such treatments [12].

The gut microbiota is different according to the anatomical regions of the gastrointestinal tract. Proteobacteria like *Enterobacteriaceae* are present in the small intestine but are absent in the colon [13]. The gut microbiota also varies with age; it typically grows from birth to adulthood and then reduces around the seventh decade of life [14].

Thus, the microbiome, through dysbiosis-induced inflammation, impaired SCFA production, and altered bile acid metabolism, contributes to T2DM progression by promoting insulin resistance, beta-cell dysfunction, and metabolic disturbances.

## 2. Methods

We conducted a literature search on PubMed, Google Scholar, and ScienceDirect using the keywords “oral antihyperglycemic drugs”, “gut microbiota and diabetes”, and “microbiome and diabetes drugs”. We manually searched all qualifying original articles by utilizing the references of the first search results, reviews, and other related publications. Since this study is a literature review, ethical clearance is not required.

The selection criteria were restricted to free full texts in English, limited to randomized clinical trials involving adults aged 19 years and older (as PubMed search criteria do not allow the selection of studies with participants over the age of 18 years). Only articles published in the last 10 years (January 2013–December 2023) were considered. Articles limited to abstracts, posters, editorials, and comments were not included in the review.

The exclusion criteria included papers with a sample size of less than 20 people over the age of 19 years and research that was not peer-reviewed. Case studies were omitted. Studies with inadequate data and those without measurable findings for outcomes were excluded.

We used a systematic review methodology based on the patient, intervention, comparison, outcome (PICO) framework developed by Eriksen and Frandsen [15].

Population: individuals aged 19 years and older who have been diagnosed with type 2 diabetes mellitus.Treatment: oral antihyperglycemic medications given to these subjects.Comparison: regular treatment vs. placebo.Objective: to determine the correlation between gut microbiota and oral antihyperglycemic medications.

The review was reported using the Preferred Reporting Items for Systematic Review and Meta-Analysis (PRISMA) (Figure 1) [16].

After careful selection, the publications that looked at oral antihyperglycemic drugs, their impact on the gut microbiota, and vice versa in drug-treated subjects or in placebo groups were selected for further analysis. For the purposes of this review, the relationship between oral drugs and the microbiome was classified into two sets: positive associations (marked with “YES”) and negative associations (marked with “NO”). “YES” associations meant that there was a direct increase in specific bacteria following the administration of certain oral antihyperglycemic drugs, and “NO” meant the opposite—decrease in specific bacteria.

A total of 8071 citations were found after searching the aforementioned databases. After removing duplicates, and other 58 articles that did not meet the search criteria, 576 were still on the list. Out of these, 287 studies were disregarded because, based on their abstracts, it was evident that they did not fit the requirements of our research; 145 papers were further dropped from consideration because they did not answer the question of this study; 78 more were excluded because access to the complete text was impossible; 13 were also omitted due to having the wrong age group; and 38 article were ignored as they were written in a language other than English. At this point, we had 15 search results that were eligible for our study.

These studies satisfied the inclusion requirements and we have systematized the data from these articles in Table 1.

Below is a breakdown of the results from the PubMed search, including the filters applied when we carried out the search.

Search results for “oral antihyperglycemic drugs”: 336 results (PubMed), 1380 results (Google Scholar), and 343 results (ScienceDirect)—total of 2059 citations.

Search results for “gut microbiota and diabetes”: 117 results (PubMed), 2145 results (Google Scholar), and 1720 results (ScienceDirect)—total of 3985 citations.

Search results for ”microbiome and diabetes drugs”: 14 results (PubMed), 1265 results (Google Scholar), and 748 results (ScienceDirect)—total of 2027 citations.

The filters applied in our search on PubMed were as follows: free full text, clinical trial, meta-analysis, randomized controlled trial, English, adult: 19+ years, and January 2013–December 2023.

The filters applied on Google Scholar were as follows: January 2013–December 2023 and any type of article.

The filters applied on ScienceDirect were as follows: January 2013–December 2023, research articles, subject areas: medicine and pharmacy, English language, and open access.

Since we could not apply additional filters for the Google Scholar and ScienceDirect database (such as open access, English language and so on), the searched citations included many results that did not apply to our current research.

## 3. Results

For this study we selected 15 studies that were analyzed and included in Table 1. We presented the main conclusions of each study and, using the PICO framework, we answered a clear and focused research question: do the antidiabetic medications alter the composition of the microbiota? The main data are provided in Table 1.

### Statistical Analysis of the Results

For the analysis of Table 1, we choose a forest plot graphical representation. This is commonly used in meta-analyses and systematic reviews to display the results of multiple studies on the same topic [65]. We created this plot to provide a visual summary of the estimated effect sizes and their confidence intervals across the 15 selected studies, allowing us to assess the overall trend and variability in the data.

The overall effect size in this forest plot is calculated as a weighted average of the individual study effect sizes. The effect size is presented with its corresponding confidence interval [66].

An effect size of 0.06 is relatively small. It is represented in the graph with a red diamond. It suggests a small difference or association between the studies included in our analysis. Since the confidence interval includes zero (the horizontal line across the red diamond shape), it suggests that the comparison of the weights of the studies may be statistically significant.

The positions of each study (the green square) are placed according to Cohen’s d, a statistical measure used to indicate the standardized difference between two means—in this case, the means of the INTERVENTION group and the CONTROL/PLACEBO group. It is particularly useful as it is calculated by taking the difference between the means of these two groups and dividing it by the pooled standard deviation. As can be seen in Figure 2, some studies [17,19,22,23,25] have a negative difference between the means of the two populations studied (calculated using Cohen’s d). This means that those who were part of the INTERVENTION group had more changes in the microbiome when compared to the CONTROL/PLACEBO group. No differentiation was made between “good” or “bad” bacteria as our analysis looked at the influence of the oral diabetic medication on the overall composition of the microbiome.

Figure 3 shows the funnel plot for the INTERVENTION and the CONTROL/PLACEBO groups, which are described in the “Comparison” section from the PICO framework.

The vertical axis measures the study precision, such as the standard error or the sample size. Studies with higher precision (larger sample size or smaller standard error) are plotted higher on the axis, while studies with lower precision are plotted lower [34]. Thus, the study with the highest accuracy in terms of sample size and standard error is the one conducted by Kondo et al. [48], with 497 individuals participating in the study, out of which 383 were diagnosed with T2DM. On the other hand, Van Bommel et al. [52] had the smallest sample size with only 44 T2DM patients, and the research showed that a 12-week treatment of Dapagliflozin or Gliclazide had no effect on the microbial composition.

To assess the relationship between the means of the INTERVENTION and CONTROL/PLACEBO groups, we ran a bivariate correlations test that helped indicate the strength and direction of this association. Confidence intervals around these correlation coefficients provide a range of plausible values for the population correlation. Table 2 shows the correlation coefficients that resulted from this test.

The numerical values are from −0.291 to 0.882, and this quantifies the strength and direction of the linear relationship between the two means. A positive correlation indicates that, as one variable increases, the other variable tends to increase as well; a negative correlation indicates that, as one variable increases, the other tends to decrease.

Table 3 shows a summary of the microbiota compositions identified in the 15 studies above and their relation to the antihyperglycemic drugs. It can be seen that Metformin treatment may lead to changes in the abundance of certain bacterial species presented below, potentially influencing the balance of beneficial and pathogenic bacteria in the gut.

Figure 4 shows a comparison between selected drugs (SGLTS2, Metformin, AGIs, Vildagliptin, Saxagliptin, and Liraglutide) and the bacteria that are being influenced by them. Metformin has a notable impact on the structure and operation of the gut microbiota. Our research compares the effects of different diabetes medications on the gut microbiome; it shows, so far, that Metformin tends to have a more pronounced impact including changes in the abundance of specific bacterial taxa and shifts in microbial metabolism. In contrast, the effects of other diabetes medications (Saxagliptin and Liraglutide) on the gut microbiome appear to be less consistent or less significant [67].

Thus, alterations in the prevalence of distinct taxa within the gut microbiota can indeed impact the equilibrium among diverse bacterial groups, leading to dysbiosis and potentially influencing various aspects of health and disease, including metabolic disorders like T2DM [68].

## 4. Discussion

### 4.1. Bacterial Phyla Commonly Found in the Gut Microbiota and Their Potential Interactions with Antidiabetic Drugs

*Firmicutes* is one of the dominant bacterial phyla in the human gut microbiota [69]. It includes various genera and species that have been implicated in glucose metabolism and insulin sensitivity [70]. For example, some studies have found correlations between an increased *Firmicutes* to *Bacteroidetes* ratio and conditions such as obesity and T2DM [70,71,72]

*Bacteroidetes* is another major phylum in the gut microbiota. Like Firmicutes, it plays a role in energy metabolism and may influence glucose homeostasis [73]. Changes in the abundance of *Bacteroidetes* have been observed in individuals with metabolic disorders, including diabetes [74]. Antidiabetic medications, such as Metformin, have been shown to modulate the ratio of *Firmicutes* to *Bacteroidetes*, which may impact glucose metabolism and insulin sensitivity [75].

*Actinobacteria* represent a smaller portion of the gut microbiota but include important genera such as *Bifidobacterium*. Certain species of *Bifidobacterium* have the ability to improve glucose tolerance and insulin sensitivity [76].

*Proteobacteria* are typically less abundant in healthy individuals, and their overgrowth has been associated with diabetes [77]. Some studies have investigated the impact of antidiabetic drugs on the composition of *Proteobacteria* in the gut, aiming to understand their role in glucose metabolism and insulin resistance [78,79,80].

*Verrucomicrobia*, similar to the aforementioned two, is a less abundant phylum in the gut microbiota, but it includes important genera like *Akkermansia muciniphila*, in particular, that has received attention for its potential beneficial effects on metabolic health, including its association with improved glucose metabolism and insulin sensitivity [81].

### 4.2. Microbial-Based Therapeutics as a Preventative Measure for T2DM

Prediabetes continues to be a stage in clinical practice that can be reversed. Probiotics have a positive impact on the body by controlling the composition of the gut flora [82]. Our research has shown that there is a clear connection between the prevalence of some bacteria and the development of diabetes. For instance, several studies have shown that probiotics may reduce insulin resistance, control blood glucose levels, reduce blood lipids, and postpone or impede the development of diabetes and its associated consequences. According to the study conducted by Pan et al., it was shown that probiotics have the ability to enhance the release of GLP-1 from L cells, resulting in a lower blood sugar level [83]. Another study conducted by Tonucci et al. showed that the consumption of probiotic fermented milk for a duration of 6 weeks improves glycemic control [84]. One the other hand, Toshimitsu et al. demonstrated an opposite effect—the efficacy of Lactobacillus plantarum OLL2712 for a duration of 12 weeks in individuals with prediabetic conditions determined improvements in fasting plasma glucose levels, glycoalbumin levels, and insulin resistance [85].

Nevertheless, the precise processes behind the impact of probiotics on prediabetes remain incompletely understood. Furthermore, there is a lack of consensus on the beneficial impacts of probiotics [86].

### 4.3. Mechanisms of Antihyperglycemic Drugs

Metformin is a medication commonly used to treat type 2 diabetes. It belongs to the biguanide class of drugs and works by reducing glucose production in the liver and increasing the body’s sensitivity to insulin [87]. Metformin distinguishes itself from other drugs by its ability to avoid the occurrence of hypoglycemia or hyperinsulinemia in individuals with T2DM [88]. It increases the uptake and utilization of glucose in the peripheral tissues [89]. The activation of adenosine monophosphate (AMP)-activated protein kinase in hepatocytes helps in the breakdown of free fatty acids [90]. In 2021, Lee et al. showed in a clinical trial that treatment with Metformin led to changes in the levels of Clostridium, Escherichia, Intestinibacter, and Romboutsia [91]. These findings are consistent with the results of our study that showed the great influence Metformin has on the microbiota [17,18,19,20,21,22,23].

Simultaneously, the increased production of SCFAs, due to the bacteria population modified by Metformin, is responsible for improving energy metabolism and restraining insulin signaling in adipose tissue [92]. Metformin intervention in obese individuals resulted in a reduction in lipopolysaccharide (LPS) production in the gut [93]. Additionally, the pool of bile acid (BA) was shown to be modified [94].

Furthermore, administering *Akkermansia muciniphila* orally to mice with high-fat-diet-induced obesity effectively improved the regulation of glucose levels and decreased inflammation in the visceral adipose tissue by stimulating the production of regulatory T cells, even without Metformin therapy [95]. This suggests that *Akkermansia* spp. may have potential as a valuable treatment for T2DM.

Sodium-glucose cotransporter 2 inhibitors upgrade glycemic regulation by promoting the elimination of glucose via the kidneys [96]. Glucosuria is caused by the inhibition of the SGLT2 cotransporter. Gliflozins prevent glucose reabsorption in the S2 segments of the proximal tubule by blocking the SGLT2 cotransporter. The improvement in glucose management is evidenced by a decrease in HbA1c levels ranging from 0.5% to 1% [97]. This is followed by other advantages, such as decreased body weight and protection against cardiovascular and renal issues [71].

In a study conducted by Elbere et al. the mice treated with Dapagliflozin showed a decrease in the abundance of *Adlercreutzia* and *Alistipes*, as well as an increase in the abundance of *Streptococcus* [98]. However, a separate clinical investigation found no notable impact on the variety or composition of microorganisms [1]. The reason for this might be because all of the participants had already received Metformin treatment, which may have influenced the potential effects of Dapagliflozin on the gut flora. Moreover, the administration of Dapagliflozin results in slight positive changes in the gut microbiota [99].

Thiazolidinedione medications may alter the balance of gut microbiota [100]. These compounds have the ability to decrease insulin resistance in adipose tissue, muscles, and the liver [101]. TZDs have the capacity to regulate glucose levels, to some extent, by decreasing free fatty acid levels [102]. Thus, in the Randle cycle, free fatty acids may be involved with glucose oxidation [103]. In mice that were given a high-fructose diet, the administration of Pioglitazone partially modified the composition of their gut microbiota [43]. This resulted in a reduction in intestinal inflammation and improvement in the integrity of the epithelial barrier. Thus, Pioglitazone may have beneficial effects on gut microbiota composition, potentially by modulating inflammation and metabolic pathways. One such effect includes the prevention of an increase in the levels of pathogenic bacteria such as *Deferribacteraceae* [104].

Dipeptidyl peptidase 4 (DDP-4) inhibitors prevent the breakdown of GLP-1 and GIP [81]. This leads to an increase in the levels of these two incretin hormones with a consequent increase in insulin production, preservation of β-cell function, and maintenance of glucose balance in the body [105,106,107]. With DPP-4 inhibited, the levels of GLP-1 and GIP in the bloodstream remain elevated for a longer period [108]. This leads to enhanced insulin secretion in response to elevated blood glucose levels, reduced glucagon release, slowed gastric emptying, and decreased appetite [109]. As shown in Table 1, experimental investigations have revealed that DPP-4 inhibitors such Saxagliptin and Vildagliptin have an effect on the gut microbiota, proved by the increased production of SCFAs in feces [110].

GLP1-Ras reduced blood sugar levels and energy intake via GLP1-receptor activation. They bind to this receptor and stimulate glucose-dependent insulin release from beta pancreatic cells and suppress glucagon release, also slowing gastric emptying and, thus, promoting satiety [78].

α-glucosidase inhibitors work by slowing down the absorption of carbohydrates in the intestines, which prevents a rapid increase in blood sugar levels after a meal [111,112,113,114]. They work primarily by targeting the enzyme α-glucosidase, which plays a crucial role in carbohydrate digestion in the small intestine, thus also having some effect on the composition of intestinal microorganisms [115,116]. Moreover, there is evidence suggesting that Sulfonylurea and Glinide may interact with probiotic bacteria or microbial metabolic profiles; however, more research is needed [117].

Over time, the gut microbiota has developed a symbiotic and mutually limiting connection with the host’s immune system and surroundings, thanks to individual adaptability and natural selection [118]. Examining the interaction mechanism between them in more detail is beneficial for comprehending individual variances in pharmacological intervention and generating insights for improving medication effectiveness, minimizing drug adverse effects, and furthering drug development [119,120,121]. The use of exceptionally stable and unique microbiota promotes the creation of microbiota preparations tailored to particular individuals and the establishment of a precision medicine system [122].

Antidiabetic medications may positively alter the gut microbiota with an improvement in overall metabolic health [123]. Several classes of antidiabetic medications have been studied in this regard, including metformin, thiazolidinediones (such as pioglitazone), and incretin-based therapies (such as glucagon-like peptide-1 receptor agonists and dipeptidyl peptidase-4 inhibitors) [63,64,86]. These medications may directly interact with gut bacteria or their metabolic pathways. For example, Metformin has been shown to accumulate in the intestine, where it can directly affect the growth and metabolism of certain bacterial species [88]. Some medications, such as DPP-4 inhibitors, modulate host physiology in ways that indirectly influence the gut microbiota. DPP-4 inhibitors work by inhibiting the degradation of incretin hormones like GLP-1, which can affect gut motility, nutrient absorption, and other factors that outline the gut environment. Alpha-glucosidase inhibitors like Acarbose delay the digestion and absorption of carbohydrates in the small intestine, leading to changes in the types and amounts of substrates available to gut bacteria [46]. These alterations in substrate availability can influence the growth and metabolism of specific bacterial species. Some antidiabetic medications have been shown to modulate immune responses, which can indirectly affect the gut microbiota. Changes in immune function can alter the gut environment and create conditions that favor the growth of certain bacterial taxa over others [61].

Conversely, SGLT2 inhibitors and TZDs have been suggested to have less pronounced effects on gut microbiota and microbial metabolites compared to other treatments [124]. Although the exact microbial patterns linked to individual antidiabetic drugs are unknown, understanding how these treatments affect the gut microbiota might be essential in identifying their potential mechanisms and improving their efficacy.

### 4.4. Limitations

One of the limitations of this study is that some of the citations included in Table 1 have a limited sample size of patients. This might result in erroneous conclusions and less precise findings. The drawbacks associated with this are reduced statistical power, a heightened error rate, and less accurate information [125].

Another challenge that we came across is that there are only a few studies carried out on humans. We found numerous titles where rats and mice were the subjects tested, but little information was available on humans.

Additionally, we found that many studies look at the rRNA gene sequencing that measures the quantity of every molecule in a cell population, as opposed to qualitative measurements. Thus, we focused on the population of bacteria rather than on the specifics of each phylum.

## 5. Conclusions

Metformin, SGLT2, GLP1-RA, DDP-4, TZD, and α-glucosidase inhibitors have been shown to have various effects on gut microbiota. Some of them increase the presence of SCFA-producing bacteria and promote its production. This may help explain why these substances are beneficial in enhancing insulin sensitivity, regulating energy metabolism, and reducing systemic inflammation. The findings of our study indicate that Metformin has a more significant influence on the gut microbiome compared to other diabetic treatments. This includes alterations in the abundance of certain bacterial taxa and changes in microbial metabolism. On the contrary, the impact of other diabetic drugs, such as Saxagliptin and Liraglutide, on the gut flora seems to display varying degrees of consistency or significance.

The modern antihyperglycemic drugs such as SGLT2 inhibitors and GLP1-receptor agonists need more human, long-duration studies regarding their interaction with intestinal flora that could reveal some other mechanisms responsible for their cardiorenal and metabolic protection.

Overall, while not the only diabetes medication that can affect the gut microbiome, Metformin appears to have a more substantial influence compared to some other drugs commonly used for diabetes management.

## Figures and Tables

**Figure 1 ijms-25-03540-f001:**
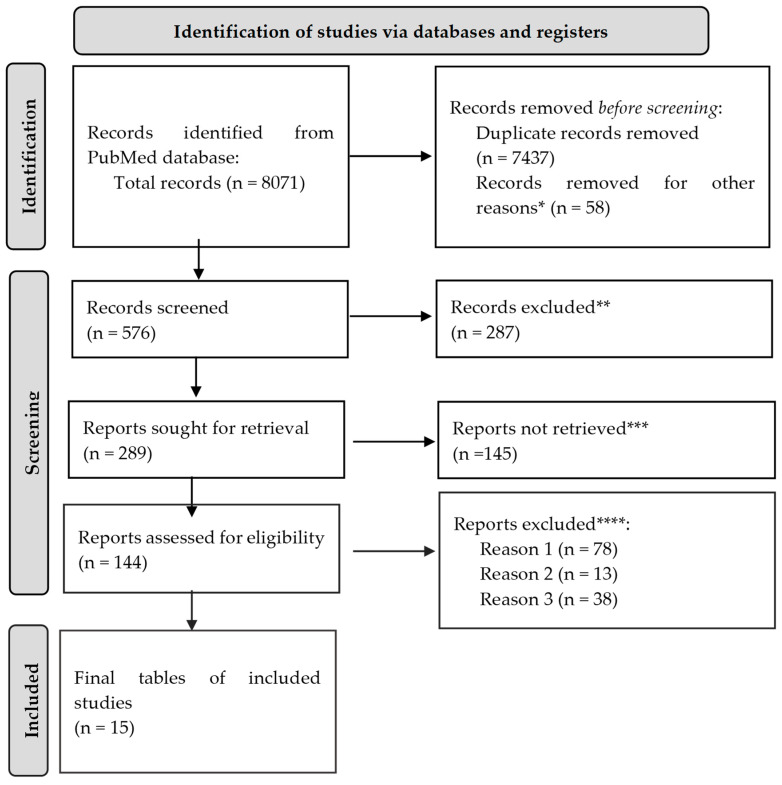
PRISMA framework. * Studies are not relevant for the present review. ** Studies do not help us to provide an answer to the current research. *** Unable to find the full text of the study. **** Reason 1—study on animals; Reason 2—wrong setting; Reason 3—research question not relevant.

**Figure 2 ijms-25-03540-f002:**
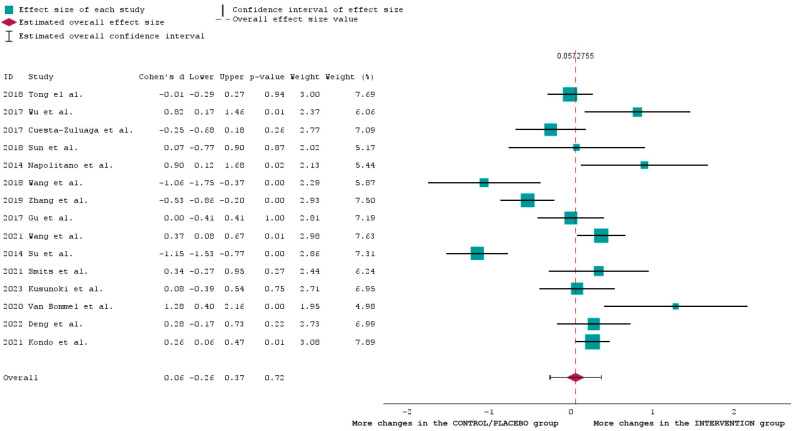
Forest plot of the 15 studies included in this research [17,20,25,32,34,36,39,44,45,48,50,52,53,59,64].

**Figure 3 ijms-25-03540-f003:**
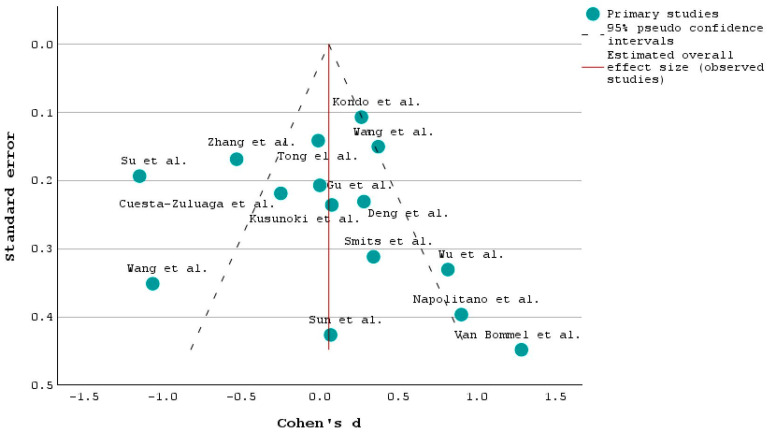
Funnel plot of the 15 studies [17,20,25,32,34,36,39,44,45,48,50,52,53,59,64].

**Figure 4 ijms-25-03540-f004:**
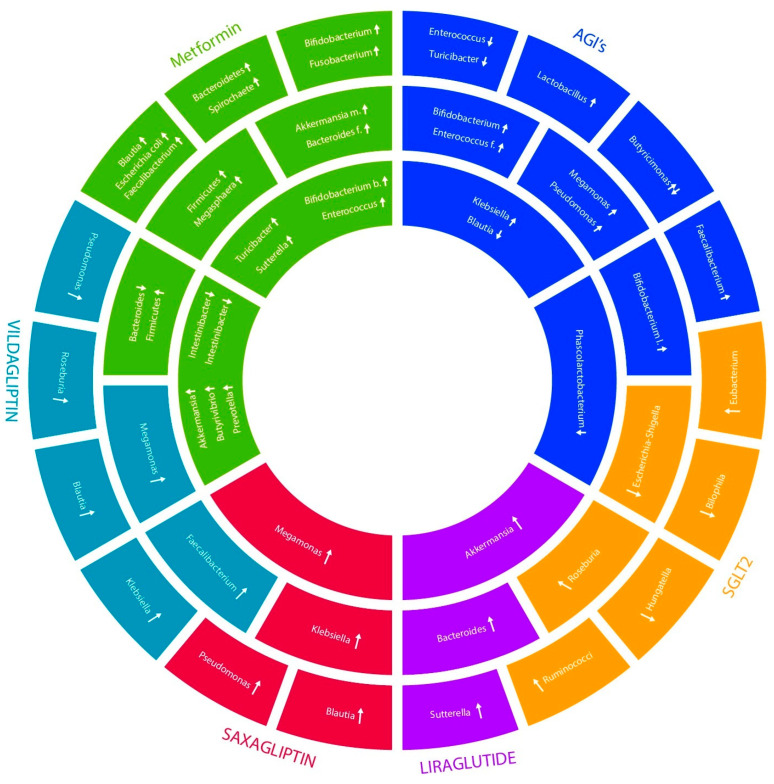
Influence of oral antidiabetic drugs on microbiome gut (↑—there is an increase in the bacteria, ↓—there is a decrease in the bacteria).

**Table 1 ijms-25-03540-t001:** Results of clinical studies investigating the impact of oral antidiabetic medications on the composition of gut bacteria in individuals with T2DM.

Antidiabetic Drugs	Study	PICO Framework	Key Results	Relation to Gut Microbiota
METFORMIN	Tong et al. [17]	Participants: 200 patients diagnosed with T2DM and hyperlipidemia. Intervention: individuals were randomized to either the Metformin-treated group or specifically designed herbal formula (AMC-treated) group. Comparison: results after 12 weeks of treatment. Outcome: the impact of the two medicines on the composition of the intestinal microbiota was assessed by analyzing the V3 and V4 regions of the 16S rRNA gene.	Both Metformin and AMC reduced high blood sugar levels and high lipid levels and caused changes in the composition of gut bacteria in individuals with diabetes. The researchers observed a substantial rise in a group of organisms called *Blautia* spp., which was strongly associated with improvements in glucose and lipid regulation. AMC demonstrated superior effectiveness in enhancing the homeostasis model assessment of insulin resistance and plasma triglyceride levels, while also showing a significant impact on gut flora. Metformin plus the AMC may improve the condition of T2DM with high levels of lipids by increasing the population of good bacteria.	YES for good bacteria: *Blautia* (regulates metabolic syndrome and inflammation) [18]; *Faecalibacterium* (reduces inflammation and promotes gut health) [19].
	Wu et al. [20]	Participants: 40 patients newly diagnosed with T2DM. Intervention: patients who had not had any medication before were randomly assigned to either receive a placebo (n = 18) or 1700 mg/d of Metformin (n = 22) for a duration of 4 months. Comparison: clinical characteristics of these individuals before and after treatment. Outcome: to identify how Metformin affects the composition of the gut microbiota.	For this study, whole-genome shotgun sequencing of 131 fecal samples was conducted. The taxonomy and gene profiles were determined by aligning the high-quality reads with nonredundant genome and gene catalogues using the metagenomic data-utilization and analysis (MEDUSA) pipeline. During the 4-month trial period, just a single bacterial strain in the placebo group underwent modification. In contrast, the administration of Metformin for 2 and 4 months led to significant changes in the prevalence of 81 and 86 bacterial species, respectively. The majority of these strains were classified as *γ-proteobacteria* (such as *Escherichia coli*) and *Firmicutes*. The results of the study also show a reduction in Intestinibacter in the group treated with Metformin.	YES for bad bacteria: *Escherichia coli* (can cause stomach cramps, bloody diarrhea, and vomiting) [21]; *Firmicutes* (increase the absorption of glucose) [22]. YES for good bacteria: *Bifidobacterium* (decrease the absorption of glucose) [23]. NO for bad bacteria: *Intestinibacter* (potentially harmful—autism) [24].
	Cuesta-Zuluaga et al. [25]	Participants: 112 individuals. Intervention: authors conducted 16S rRNA gene sequencing to examine the formation and arrangement of the gut microbiota. Comparison: 28 T2DM individuals, with 14 of them using Metformin and 84 individuals without diabetes who were selected to match the participants with diabetes in terms of sex, age, and BMI at a ratio of 3 to 1. Outcome: to find out if Metformin is linked to high levels of bacteria that produce short-chain fatty acids and degrade mucin.	A link was discovered between diabetes and gut microbiota, which was influenced by the usage of Metformin. Participants with diabetes who were taking Metformin had a greater occurrence of Akkermansia muciniphila, a type of microbiota that is known for breaking down mucin, as well as several types of gut microbiota that are known for producing SCFAs, including *Butyrivibrio*, *Bifidobacterium bifidum*, *Megasphaera*, and a specific group within the *Prevotella* taxonomic unit. People with diabetes who were not taking Metformin showed a greater frequency of Clostridiaceae 02d06 and a unique operational taxonomic unit of *Prevotella*, as well as a reduced abundance of *Enterococcus casseliflavus*, in comparison to people without diabetes.	YES for good bacteria: *Akkermansia* (protective effect in obesity, diabetes, and inflammation) [26]; *Butyrivibrio* (inversely associated with obesity) [27]; *Bifidobacterium bifidum* (produces B vitamins and healthy fatty acids) [28]; *Megasphaera* (possible role in achieving a healthy gut) [29]; *Prevotella* (biomarker for the development of diabetes) [30]. NO for bad bacteria: *Enterococcus casseliflavus* (causes urinary tract and abdominal infections) [31].
	Sun et al. [32]	Participants: 22 T2DM patients. *Intervention:* serum and stool samples were collected from the individuals with T2D. Comparison: the microbiota of the participants was analyzed before and after being treated with 1000 mg Metformin twice daily for 3 days. Outcome: to investigate how Metformin controls gut microbiota and metabolites in humans.	The abundance of *Bacteroides fragilis* was reduced, but the concentration of the bile acid glycoursodeoxycholic acid was higher in the gastrointestinal tract. The alterations were accompanied by the suppression of intestinal farnesoid X receptor signaling. Metformin functions, at least partially, by using a B. fragilis–GUDCA–intestinal FXR axis to enhance metabolic dysfunction, such as in hyperglycemia.	YES for good bacteria: *Bacteroides fragilis* (essential to mucosal immunity) [33].
	Napolitano et al. [34]	Participants: 14 T2DM patients. Intervention: all subjects had to be on a stable dose of Metformin of ≥1000 mg/day for more than 3 months, which was stopped and later resumed. Comparison: subjects were studied at 4 time points: (i) at baseline on Metformin; (ii) 7 days after stopping it; (iii) when fasting blood glucose (FBG) had risen by 25% after stopping Metformin; (iv) when FBG returned to baseline levels after restarting Metformin. Outcome: to characterize the gut-based mechanisms of Metformin.	Discontinuing Metformin led to a decrease in both active and total GLP-1 levels, while causing an increase in serum bile acids, particularly cholic acid and its conjugates. The aforementioned effects were reversed with the resumption of Metformin. The impact on circulating PYY was rather small, but the alterations in GIP were insignificant. The firmicutes phylum microbiota was positively linked with changes in cholic acid. On the other hand, the *Bacteroidetes phylum* microbiota was negatively correlated with it. The presence of *Firmicutes* and *Bacteroidetes* in the gut microbiota was shown to be associated with the levels of serum PYY. Thus, Metformin has intricate effects resulting from its pharmacological actions in the gut.	YES for bad bacteria: *Firmicutes*. NO for bad bacteria: *Bacteroidetes* (influence on glucose and fat metabolism) [35].
	Wang et al. [36]	Participants: 37 T2DM patients. Intervention: Part B subjects were switched from oral Metformin to subcutaneous once daily injections of Liraglutide began. Part C subjects remained on Metformin. Comparison: the subjects who were stable on Metformin were randomized into two study arms—Part B (n = 19) and Part C (n = 18). Part A comprised only health volunteers. Outcome: to analyze, after 42 days of trial, the effects of these drugs on the composition of the microbiome.	Both before and after the trial, individuals who were taking Metformin experienced a rise in the proportion of the bacterial group Sutterella. Also, Liraglutide had a positive association, leading to an increase in the bacterial group *Akkermansia*. The relative abundances of *Bacteroides* and *Akkermansia* were strongly linked to the duration of diabetes in the subjects. More precisely, those with shorter and medium durations of diabetes had a notably greater prevalence of *Akkermansia* compared to those with a longer duration of the condition.	YES for bad bacteria: *Sutterella* (associated with autism and inflammatory bowel disease) [37]. YES for good bacteria: *Akkermansia*; *Bacteroides* (decrease the absorption of glucose) [38].
	Zhang et al. [39]	Participants: 180 individuals with and without T2DM. Intervention: microbiome compositions were analyzed via a 16S ribosomal RNA gene-based sequencing protocol. Comparison: 130 T2DM patients with a specific hypoglycemic treatment and 50 healthy volunteers. Outcome: to identify how the diabetes treatment affects the microbiota.	The use of hypoglycemic drugs resulted in changes to certain species within the gut microbiota, rather than affecting its overall diversity. Metformin boosted the prevalence of *Spirochaete*, *Turicibacter*, and *Fusobacterium*. Insulin further raised the levels of *Fusobacterium*, whereas α-glucosidase inhibitors (α-GIs) were responsible for the abundance of *Bifidobacterium* and *Lactobacillus*. Both Metformin and insulin improved the metabolism of taurine and hypotaurine, whereas α-GI stimulated many amino acid pathways. While there were similarities in the gut microbial community across those using Metformin and insulin, there were notable differences in each diabetic group with hypoglycemia.	YES for bad bacteria: *Spirochaete* (associated with Syphilis, Lyme disease, and Leptospirosis) [40]; *Turicibacter* (weight gain and changes in the serum levels of total triglycerides and total cholesterol) [41]. YES for good bacteria: *Fusobacterium* (produces lipopolysaccharides responsible for the production of cytokines and other inflammatory mediators) [42]; *Bifidobacterium*; *Lactobacillus* (improves cardiovascular diseases, lactose intolerance, prevents and treats cancer, and regulates immunity) [43].
AGIs and SULFONY-LUREAS	Gu et al. [44]	Participants: 94 treatment-naïve T2DM patients. Intervention: at the start and after 3 months of therapy, samples of feces and blood were collected. Comparison: 1:1 randomized into Acarbose and Glipizide groups. Outcome: to characterize the clinical effects of Acarbose and Glipizide.	After 3-month therapy, there were substantial decreases in HbA1c levels, as well as fasting and postprandial blood glucose levels, in both groups. The Acarbose group showed a higher decrease in body weight and BMI compared to the Glipizide group. Patients who were administered Acarbose, but not Glipizide, displayed a significant improvement in clinical parameters that are risk factors for metabolic comorbidities and cardiovascular complications associated with T2DM (homeostasis model assessment of insulin resistance, total cholesterol, triglyceride levels, and fatty liver index). Both Acarbose and Glipizide therapy resulted in a reduction in plasma FGF19 levels (a crucial factor generated in the gut that plays a significant role in metabolic health). This suggests that the improvement in HbA1c, FBG, and PBG levels was not subject to FGF19.	NO effect on microbial composition.
	Su et al. [45]	Participants: 140 participants with and without T2DM. Intervention: inflammatory cytokines were determined using either ELISA or RT-PCR. Comparison: 59 participants were assigned to Group A, who received antidiabetic medication (150 mg of Acarbose per day), and 36 participants to Group B (no Acarbose but received the same treatment as Group A). The control group was formed of 45 healthy individuals. Outcome: to analyze trend differences between the two diabetic groups.	After a 4-week treatment, *Bifidobacterium longum* and *Enterococcus faecalis* were seen to have grown in both diabetic groups. Group A had a greater abundance of *Bifidobacterium longum*, along with reduced levels of LPS and prothrombin activator inhibitor-1. *Enterococcus faecalis* had a negative association with LPS, whereas *Bifidobacterium longum* presented a favorable connection with Acarbose treatment and HDL cholesterol levels. Acarbose treatment may increase the presence of *Bifidobacterium longum* in the intestines of people with type 2 diabetes as well as decrease some inflammatory cytokines, independent of its ability to lower blood sugar levels.	YES for good bacteria: *Bifidobacterium longum* (produces lactic and acetic acid in the gut) [46]. YES for bad bacteria: *Enterococcus faecalis* (can cause infection when it enters the body via a wound, blood, or urine) [47].
	Kondo et al. [48]	Participants: 497 individuals. Intervention: collection of fecal samples and analysis the makeup of gut bacteria were conducted. Comparison: 383 patients with T2DM and 114 individuals without T2DM were classified into red, blue, green, and yellow groups. Outcome: to compare the proportions of phyla and genera following the grouping of the gut microbiota into four distinct groups.	The red group had higher proportions of the *Bifidobacterium* and *Lactobacillus* genera, while demonstrating reduced proportions of the *Blautia* and *Phascolarctobacterium* genera. The red group had a greater percentage of individuals with T2DM who used α-glucosidase inhibitors and Glinide medicines and had a reduced consumption of fermented soybean foods, such as miso soup, compared to the other groups. Additionally, these findings indicate that certain medications for diabetes and fermented food products may play a role in this alteration.	YES for good bacteria: *Bifidobacterium*; *Lactobacillus*. NO for good bacteria: *Blautia*; *Phascolarctobacterium* (positively correlated with alpha-linolenic acid and decreases the risk of heart disease) [49].
SGLT2 INHIBITORS	Kusunoki et al. [50]	Participants: 36 patients with T2DM. Intervention: individuals received a SGLT2 inhibitor (Luseogliflozin or Dapagliflozin) for 3 months. Comparison: the presence of germs in the feces of the patients was assessed both before and after treatment with SGLT2 inhibitors. Outcome: to evaluate the incidence rates of microorganisms that regulate and maintain the equilibrium of the microbiota.	Treatment with SGLT2 inhibitors was shown to significantly enhance the total prevalence of the 12 species of bacteria that regulate balance. Furthermore, there were notable increases in the occurrences of bacteria that produce SCFAs among the microorganisms responsible for maintaining balance. Specific examination of the bacteria responsible for maintaining balance in the body showed that treatment with the SGLT2 inhibitor resulted in a notable rise in the occurrence of *Ruminococci*. Nevertheless, the SGLT2 inhibitor did not have any impact on the bacteria that disrupt the equilibrium. Thus, SGLT2 inhibitors are linked to a rise in the occurrence of bacteria that regulate balance.	YES for good bacteria: *Ruminococci* (breaks down dietary fiber and promotes gut balance) [51].
	Van Bommel et al. [52]	Participants: 44 T2DM patients. Intervention: 16S rRNA gene sequencing was used to assess the microbiome. Comparison: for 3 months, 44 patients were randomized to either Dapagliflozin or Gliclazide treatment. Outcome: the microbiome of patients who were already receiving Metformin therapy was analyzed after they received either Dapagliflozin or Gliclazide.	Although both Dapagliflozin and Gliclazide improved glycemic management, Dapagliflozin decreased fasting insulin levels and Gliclazide raised them. Dapagliflozin significantly improved the excretion of glucose in urine; however, Gliclazide did not. Dapagliflozin also led to a reduction in BMI, fat mass percentage, and waist circumference, whereas Gliclazide increased them. However, both treatments had no significant impact on either the diversity or composition of the microbiota.	NO effect on microbial composition.
	Deng et al. [53]	Participants: 76 treatment-naïve patients with T2DM at risk of cardiovascular diseases (CVDs). Intervention: patients were treated with either Empagliflozin (10 mg/d, n = 40) or Metformin (1700 mg/d, n = 36). Comparison: the clinical parameters of the two groups were compared. Outcome: to evaluate the changes related to glucose metabolism, CVD’s factors, and gut microbiota using 16S rRNA gene sequencing and plasma metabolites.	HbA1c levels decreased in both groups, but only Empagliflozin group showed a change in the microbiome and an increase in CVD risk. The same group showed raised plasma metabolite levels, while having decreasing levels of glycochenodeoxycholate, cis-aconitate, and uric acid. Simultaneously, Empagliflozin increased the abundance of species from Roseburia, Eubacterium, and *Faecalibacterium* (SCFAs producing bacteria), while decreasing the presence of many hazardous bacteria, including *Escherichia-Shigella*, *Bilophila*, and *Hungatella*.	YES for good bacteria: *Roseburia* (butyrate production and protective role in most digestive diseases) [54]; *Eubacterium* (produces butyrate and plays a critical role in energy homeostasis, colonic motility, immunomodulation, and the suppression of inflammation in the gut) [55]; *Faecalibacterium*. NO for bad bacteria: *Escherichia-Shigella* (invasion and inflammatory destruction of the human colonic epithelium) [56]; *Bilophila* (produces hydrogen sulfide that breaks down the intestinal wall and enhances the progression of inflammatory bowel disease) [57]; *Hungatella* (association with severe diseases, sporadic cases of bacteremia, fatal septicemia, and severe cases of COVID-19) [58].
DDP-4 INHIBITORS	Wang et al. [59]	Participants: 90 T2DM patients. Intervention: individuals were treated with Acarbose, Saxagliptin, and Vildagliptin. Comparison: groups of 30 patients for each medicine. Outcome: to evaluate the efficacy of Acarbose, Saxagliptin, and Vildagliptin in the treatment of T2DM.	Patients had examinations at 0, 4, and 12 weeks post-treatment, during which their vital signs were documented. Fecal samples were collected for the purpose of conducting microbial macrogenome sequencing and safety assessments. There was a reduction in blood glucose levels at 4 and 12 weeks after therapy, and there was a notable difference in the total cholesterol and HDL levels at the 12-week mark. Acarbose first raised the amount of *Butyricimonas* but then reduced it throughout the course of medication administration. Saxagliptin caused a progressive rise in the level of the *Megamonas* genus. Also, it reduced the level of the *Turicibacter* genus. The levels of *Pseudomonas*, *Klebsiella*, *Blautia*, *Faecalibacterium*, and *Roseburia* varied during Vildagliptin administration, resulting in a larger rise in fasting C-peptide levels compared to the other two medications. Saxagliptin displayed a higher incidence of adverse events compared to Acarbose and Vildagliptin. The combined use of the three medicines may significantly lower the HbA1c level and impact the distribution of intestinal flora in individuals with T2DM.	YES for good bacteria: *Butyricimonas*—increased then decreased (beneficial effect on metabolic disorders) [60]; *Faecalibacterium*. YES for bad bacteria: *Megamonas* (abdominal fat weight and ratio) [61]; *Pseudomonas* (can cause meningitis, otitis media, urinary tract infections, and pneumonia) [62]; *Klebsiella* (can cause pneumonia, bloodstream infections, wound or surgical site infections, and meningitis) [63]; *Blautia*. NO for bad bacteria: *Turicibacter*.
	Smits et al. [64]	Participants: 51 patients with T2DM. Intervention: individuals received, once a day for 12 weeks, either Liraglutide, Sitagliptin, or placebos. Comparison: fecal samples were analyzed using 16S rRNA gene sequencing at baseline and after 12 weeks. Outcome: to evaluate the impact of Liraglutide, Sitagliptin, or placebos on the composition of the gut microbiota.	Patients who were already taking Metformin or Sulphonylureas were given either Liraglutide or Sitagliptin. When taken as an additional treatment in T2DM patients who are already taking Metformin, the conclusion is that it does not significantly change the composition of the gut microbiota compared to a placebo.	NO effect on microbial composition.

**Table 2 ijms-25-03540-t002:** Bivariate correlations with confidence intervals.

Intervention	Control	Correlation	Count	Lower C.I.*	Upper C.I.
Mean	Mean	0.997	15	0.990	0.999
	SD	0.692	15	0.279	0.889
	Population	−0.151	15	−0.616	0.391
SD*	Mean	0.770	15	0.426	0.920
	SD	0.882	15	0.675	0.960
	Population	0.044	15	−0.479	0.544
Population	Mean	−0.291	15	−0.699	0.260
	SD	−0.233	15	−0.666	0.317
	Population	0.714	15	0.318	0.898

SD* = standard deviation; C.I.* = confidence interval.

**Table 3 ijms-25-03540-t003:** Antihyperglycemic drugs that increase the bacteria in the gut (↑—there is an increase in the bacteria, ↓—there is a decrease in the bacteria).

Bacteria in the Gut	Antihyperglycemic Drugs		
*Blautia*	METFORMIN ↑		
*Faecalibacterium*	METFORMIN ↑	ACARBOSE ↑	VILDAGLIPTIN ↑
*Escherichia coli*	METFORMIN ↑		
*Firmicutes*	METFORMIN ↑		
*Bifidobacterium*	METFORMIN ↑	ACARBOSE ↑	
*Intestinibacter*	METFORMIN ↓		
*Akkermansia m.*	METFORMIN ↑		
*Butyrivibrio*	METFORMIN ↑		
*Bifidobacterium b.*	METFORMIN ↑		
*Megasphaera*	METFORMIN ↑		
*Prevotella*	METFORMIN ↑		
*Enterococcus casseliflavus*	METFORMIN ↓	ACARBOSE ↑	
*Bacteroides f.*	METFORMIN ↑		
*Firmicutes*	METFORMIN ↑		
*Bacteroidetes*	METFORMIN ↓		
*Sutterella*	METFORMIN ↑	LIRAGLUTIDE ↑	
*Akkermansia*	METFORMIN ↑	LIRAGLUTIDE ↑	
*Bacteroides*	METFORMIN ↑	LIRAGLUTIDE ↑	
*Spirochaete*	METFORMIN ↑		
*Turicibacter*	METFORMIN ↑	ACARBOSE ↓	
*Fusobacterium*	METFORMIN ↑		
*Bifidobacterium*	METFORMIN ↑	AGIs ↑	
*Lactobacillus*	METFORMIN ↑	AGIs ↑	
*Butyricimonas*		ACARBOSE ↑↓	
*Megamonas*	SAXAGLIPTIN ↑	ACARBOSE ↑	VILDAGLIPTIN ↑
*Pseudomonas*	SAXAGLIPTIN ↑	ACARBOSE ↑	VILDAGLIPTIN ↑
*Klebsiella*	SAXAGLIPTIN ↑	ACARBOSE ↑	VILDAGLIPTIN ↑
*Blautia*	SAXAGLIPTIN ↑	ACARBOSE ↓	VILDAGLIPTIN ↑
*Phascolarctobacterium*		AGIs ↓	
*Bilophila*	SGLT2 ↓		
*Hungatella*	SGLT2 ↓		
*Bifidobacterium l.*		ACARBOSE ↑	
*Enterococcus f.*		AGIs ↑	
*Ruminococci*	SGLT2 ↑		
*Roseburia*	SGLT2 ↑		VILDAGLIPTIN ↑
*Eubacterium*	SGLT2 ↑		
*Escherichia-Shigella*	SGLT2 ↓		

## Data Availability

No new data were created or analyzed in this study. Data sharing is not applicable to this article.

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
