# Peer review of "Interactions between Gut Microbiota and Oral Antihyperglycemic Drugs: A Systematic Review"

_ijms, 2024, doi:10.3390/ijms25063540_

Round 1
Reviewer 1 Report
Comments and Suggestions for Authors
In this systematic review, the authors highlight current update knowledge on the crosstalk and interactions between gut microbiota and antidiabetic drugs, including metformin, α-glucosidase inhibitors, dipeptidyl peptidase-4 inhibitors, and sodium-glucose cotransporter 2 inhibitors. On the 15 final selected studies, the authors concluded that all the antidiabetic drugs have demonstrated similar effect to metformin with regard to beneficial modulation of gut microbiota. The authors discussed potential study limitations.
Minor comments:
1. In Fig. 5, authors should include potential mechanisms by which diabetic medication influences gut microbiota.
2. Authors should include this relevant study:
Liu W, Luo Z, Zhou J, Sun B. Gut Microbiota and Antidiabetic Drugs: Perspectives of Personalized Treatment in Type 2 Diabetes Mellitus. Front Cell Infect Microbiol. 2022 May 31;12:853771. doi: 10.3389/fcimb.2022.853771. PMID: 35711668; PMCID: PMC9194476.
Comments on the Quality of English LanguageCheck some few typos.
The manuscript is well written and content may be of great interest to readers and researchers in the field of endocrinology.
Reviewer 2 Report
Comments and Suggestions for Authors
The relevance of the study is beyond doubt; the very topic of the influence of modern antihyperglycemic drugs on the microbiome is extremely interesting and in demand, at the same time complex and diverse.
The strength of the study is the statistics; they are described in detail and, as far as I can tell, correctly applied.
But there are a number of comments and questions:
The following conclusion is not entirely clear: Metformin, SGLT2, DDP-4 and α-glucosidase inhibitors have been shown to have similar effects on gut microbiota. They increase the presence of SCFA-producing bacteria and promote SCFA production. (Lines 419, 420). I consider it incorrect.
Taking a detailed look at the results, we see that when taking DPP-4, the level of conditionally bad bacteria Pseudomonas, Klebsiella increased, when taking metformin, the number of good bacteria (Blautia, Faecalibacterium, Bifidobacterium, Lactobacillus) increased, and you conclude that their effect on the microbiome is similar ?
In fact, several times in the text I see statements about the positive effect of vildagliptin and saxagliptin on the microbiome, but I don’t see numbers confirming this. Is an increase in the level of pathogenic bacteria really a positive effect?
I think the discussion is insufficient - it contains many general phrases that do not explain the essence of the study, but somewhat blur it. I find the discussion and results difficult to understand, and perhaps the interpretation of the results is not entirely accurate. As a result, the article is largely incomprehensible to the reader. I would like a little more information about the bacteria themselves.
Apparently erroneous reference to Table 2: “As shown in Table 2, experimental studies have shown that DPP-4 inhibitors such as saxagliptin and vildagliptin have the potential to promote energy metabolism by influencing the composition of gut microbes and increasing the production of SCFAs 366 in feces [ 85]. At the same time, “Table 2. Bivariate correlations with confidence intervals.”
I think it is premature to conclude that microbiota can be used to treat type 2 diabetes (data from a small sample), I don’t think it’s worth including them in the abstract (lines 29, 30, 31). Or at least make the statement less categorical. This conclusion needs to be confirmed by other studies.
I would like to positively note the list of analyzed literature, the vast majority of the sources (75%) are no older than 5 years, and many of them are even from 2023; many studies have been analyzed, which certainly indicates that a lot of work has been done.
Reviewer 3 Report
Comments and Suggestions for Authors
This manuscript by Mindrescu et al explores the relationship between the gut microbiome and anti-hyperglycemic drugs. The authors referenced several recent studies where antihyperglycemic drugs and their effects on gut microbiome were studied. Overall, the manuscript is well written, has a specific aim. Below are a few comments:
1. Mechanisms of antihyperglycemic drugs: It would be beneficial to discuss the mechanism of action of the different categories of antihyperglycemic drugs with some more detail. As such, it is briefly mentioned under the discussion section, but a little more detail will be a great addition.
2. The abstract mentions that the review also discusses microbial based therapeutics for use as a preventative measure for type 2 Diabetes mellitus, however, this has not been discussed at any point in the manuscript
3. The conclusion needs to have more details from the different findings of the review
4. Please make sure all abbreviations have been defined in their respective first appearance in the text. For example, no definition for DPP-4, etc
Comments on the Quality of English LanguageManuscript has some spelling and grammatical errors
